# Delay in the Ripening of Wine Grapes: Effects of Specific Phytotechnical Methods on Harvest Parameters

**Gizella Jahnke** [1,*] **, Barna Árpád Szőke** [1] **, Szabina Steckl** [2] **, Áron Pál Szövényi** [2] **, Gyöngyi Knolmajerné Szigeti** [1] **, Csaba Németh** [1] **, Botond Gyula Jenei** [1] **and Diána Ágnes Nyitrainé Sárdy** [2]

1   Badacsony Research Station, Institute for Viticulture and Oenology, Hungarian University of Agriculture and Life Sciences, 8261 Badacsonytomaj, Hungary; szoke.barna.arpad@uni-mate.hu (B.Á.S.); knolmajerne.szigeti.gyongyi@uni-mate.hu (G.K.S.); nemeth.csaba@uni-mate.hu (C.N.); jenei.botond.gyula@stud.uni-mate.hu (B.G.J.)

2   Department of Oenology, Institute for Viticulture and Oenology, Hungarian University of Agriculture and Life Sciences, 118 Budapest, Hungary; steckl.szabina@uni-mate.hu (S.S.); szovenyi.aron.pal@uni-mate.hu (Á.P.S.); nyitraine.sardy.diana.agnes@uni-mate.hu (D.Á.N.S.)

*   Correspondence: gyorffyne.jahnke.gizella@uni-mate.hu

**Abstract:** Due to climate change, the sugar content of grapes in Hungary has increased to such an extent that the high alcohol content alone can make wines disharmonious. In most vintages, this phenomenon is only a problem for early-ripening varieties. In order to prevent and treat this difficulty, we have carried out experiments in grape canopy management for four years with the aim of delaying ripening and thus reducing the sugar content of the grapes. The experiments were set up on an early (Pinot noir) and a late (Welshriesling) variety; two treatments (leaf removal—LR and short topping—ST) were applied and compared to untreated controls in the years 2019–2022. Our results showed that grape juice sugar yield was significantly reduced in all four years and for both cultivars, while the other measured parameters (yield, acidity, pH, and Botrytis infection) were only lightly affected.

**Keywords:** DMR; harvest time; *Vitis vinifera*; climate change; global warming

## 1. Introduction

Climate change has two effects that have a significant impact on viticulture. These are changes in temperature and rainfall. According to the latest IPCC report [1], even the most optimistic projections suggest that vine-growing areas could see a minimum annual increase in average temperatures of 1–1.5 °C (Figure 1). Annual precipitation will increase in some areas and decrease in others but the annual distribution of precipitation will in any case change unfavorably so that in most wine-growing areas there will be a shortfall in precipitation during the growing season (Figure 2).

Global warming has a negative impact on the quality of white wines, mainly due to the loss of acidity and the lack of aromatic ripeness caused by too-rapid ripening. Due to the milder winters, there will be greater pest and disease pressure [2,3]. Hot summers result in earlier grape ripening and in some wine-growing regions diseases such as Botrytis are more likely to appear [4,5]. The increase in ultraviolet-B (UV-B) radiation on the soil surface due to the decreased ozone layer can cause changes in the physiology of the vine and have a direct effect on grape composition. The aromatic profiles may change and the aroma of white wine varieties in particular may be less marked [2].

The minimal thermal demand for grapevine growth is expressed as a value of the heat summation index (growing degree-days [GDD] from April to October in the northern hemisphere, with a base temperature of 10 °C). Becker et al. [6] specified the minimum GDD as 1000 (°D units); however, subsequent research has found the minimum to be



850 [7–9]. In the last decade, the vine development phases, such as budburst, bloom, and harvest have, on average, taken place earlier than in the 1980s [10–14].

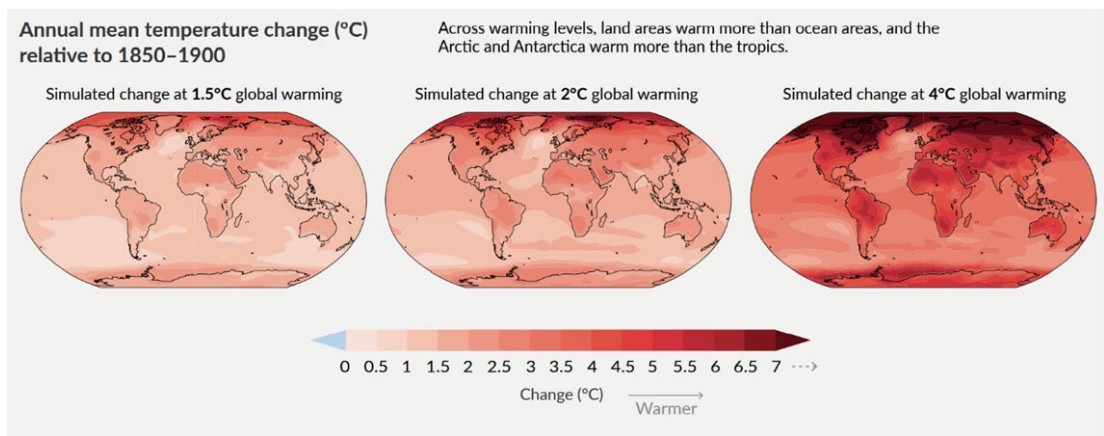

**Figure 1.** Annual mean temperature change (°C) relative to 1850–1900 according to Shukla et al. (2022) [1].

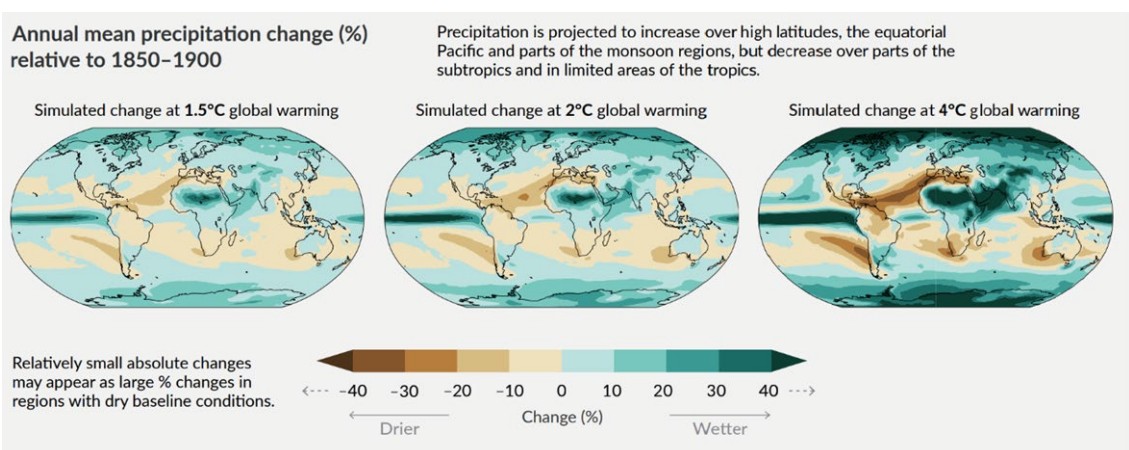

**Figure 2.** Annual mean precipitation change (%) relative to 1850–1900 according to Shukla et al. (2022) [1].

According to Van Leeuwen et al. [15], the suitability for winegrowing in the world's most important wine-producing regions will not decline significantly over the next four decades. They identify significant methodological flaws in the article by Hannah et al. [16]—the alarming statement is primarily related to (i) the misuse of bibliographical data to compute suitability index, (ii) the underestimation of adaptations of viticulture to warmer conditions, and (iii) the inadequacy of the monthly timestep in the suitability approach. Van Leeuwen et al. also gave some examples about the adaptation of wine growing in Rheingau, (Germany) and in Burgundy and the Rhone Valley (France)—Figure 3.

Hannah et al. [17] replied that climate change adaptation has started in vineyards but the way in which the wine industry develops in the future decades will affect wildlife. Dry farming may be an early response but planning and study are required to keep up with increasing temperatures. When planning agricultural climate change solutions, ecosystem services, wildlife, and water ought to be included [17].

Using the bias-corrected outputs of three distinct regional climate models (RegCM, ALADIN, and PRECIS), the spatial distribution of key indicators describing wine production in Hungary was examined. The daily minimum, maximum, and mean temperature and daily precipitation time series were used for this purpose. In this research, the previous

changes in these indices were analyzed first and then the anticipated changes until the end of the 21st century were the primary emphasis [18]. When calculating the most important climate indicators used in viticulture (e.g., Huglin index), it is important to know the length of the growing season (more precisely, the beginning and the end). Mesterházy et al. [19] proposed to calculate the length of the growing season on the basis of temperature instead of the previously widely used period from 1 April to 30 September. The essence of their method was to take the middle day of the first and last five-day period with a daily mean temperature of at least 10 °C as the beginning and end of the growing season, respectively [20]. This method can be used to refine our estimates and conclusions for the future. The possible loss of supremacy of white wine grapes over red wine, as well as the increase in the importance of late- and very-late-ripening grape types in Hungary in the next decades, was projected. Authors also suggest the increase in the frequency of very high summer temperatures and the decrease in the danger of frost damage throughout the reproductive cycle [20].

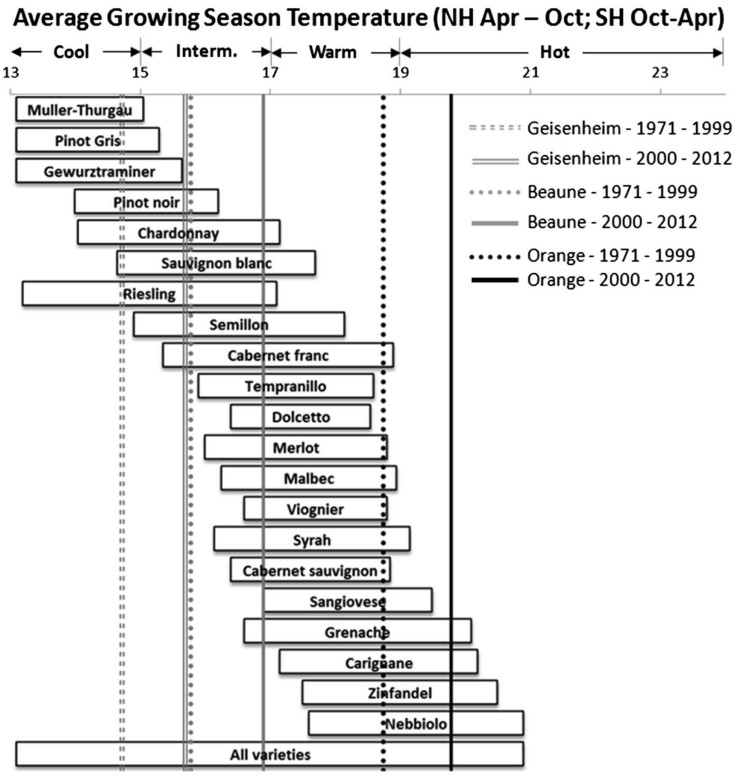

**Figure 3.** Average growing season temperature from 1971 to 1999 and from 2000 to 2012 in Rheingau, Germany (Geisenheim station, Deutscher Wetterdienst); Burgundy, France (Beaune station); and Rhone Valley, France (Orange station)—according to van Leeuwen et al. (2013) [15].

Research was conducted in 2006 at the Ampelographic Collection of the Horticulture Faculty in Iasi on the Zweigelt variety [21]. The effect of the total leaf area, canopy thickness, and direct sun radiation on crop quality was analyzed. The relationships between the canopy parameters and crop quality were determined. The total foliage area was shown to have a positive correlation with sugar content in must, alcohol concentration in wine, total extract, and total acidity. The anthocyan content of grapes and wine decreases as the thickness of the canopy increases and as the foliage's exposure to direct sun radiation decreases. According to this study, the adjustment of canopy parameters altered the anthocyanic profile and the chromatic features of the wines.

The study conducted at a commercial vineyard in Brazil was to assess the influence of canopy management on the composition of Sauvignon blanc grapes. During the 2005/2006 and 2006/2007 seasons, interventions for canopy control were implemented by topping

shoots. From véraison until harvest, ripening was assessed weekly. It was found that the leaf area treatments influenced the berry accumulation of soluble solids and titratable acidity but had a minimal effect on other factors [22].

The orientation of the rows, the exposure of the canopy, and the ripeness of the grapes all contribute to the sensory characteristics of wine. The objective of the research of Minnaar et al. [23] was to determine the influence of canopy exposure on selected sensory characteristics of Pinotage and Cabernet Sauvignon wines from Paarl, Durbanville, and Darling in South Africa. The east side of Durbanville Cabernet Sauvignon wines have enhanced color, aroma, mouthfeel, and overall quality. The south side of Paarl Cabernet Sauvignon wines has improved color, aroma, mouthfeel, and overall quality. West-side Darling Pinotage wines showed enhanced aroma and acidity intensity while east-side Durbanville Pinotage wines had a higher alcohol, pH, TA, color, and aroma intensity, as well as overall quality. These studies demonstrate that canopy exposure influences the sensory characteristics of wine.

Grape cluster positions affect sunlight and grape berry compounds. Gao et al. [24] examined how cluster positions in the canopy (interior and two exterior canopy sides) affected flavonoid and volatile compound profiles of *Vitis vinifera* L. cultivars Cabernet franc and Chardonnay berries in two consecutive years. Clusters within the canopy received less sunshine than those outside and their average temperatures changed somewhat. Throughout two years, cluster placements in the canopy did not affect the cluster weight, berry weight, juice total soluble solids, or titratable acidity for either cultivar. The inner clusters of both cultivars showed lower total flavanol contents than the exterior clusters but the canopy location did not affect the anthocyanin or flavan-3-ol composition. The position of clusters affected volatile chemicals and certain bound norisoprenoids and terpenoids were lower in inner clusters than in outer clusters.

The primary purpose of the research by Prezman et al. [25] was to reduce the alcohol concentration of wine by using a combination of procedures from the vineyard to the cellar. The combination of these procedures should result in a 2% volume reduction in wine's alcohol content. 'Tannat' N and 'Gros Manseng' B, two of the most important grape varieties in the southwest of France, were the subject of a two-year experiment. Nowadays, in the context of climate change, grapes often produce up to or more than 15% of potential alcohol. In order to delay ripening and produce more digestable wines, three cultural strategies were evaluated and compared to the control: leaf removal on the top canopy, canopy reduction by late hedging, and anti-transpirant spraying on the whole canopy. Using yeast with a low alcoholic output, these methods were paired with a biological process to decrease the alcohol production. Both low-yield *Saccharomyces cerevisiae* yeast and control yeast were used to vinify four replicates. Findings indicated that late hedging was the most effective method for delaying ripening in both cultivars but it also had an effect on characteristics like as acidity and polyphenols. Other evaluated viticultural practices were similarly effective in slowing down ripening. Low alcoholic yield yeast results in lower alcohol concentration, more acidic wines, and less volatile acidity during winemaking.

Gambacorta et al. aimed to determine the effect of early basal leaf removal on Aglianico wines produced in Apulia (southern Italy) over three consecutive growing seasons. In each of the three treatments, all of the cluster-zone leaves on the north, south, and both sides of the canopy were removed. Early defoliation enhanced the levels of flavonoids (+40%), anthocyanins (+18%), total polyphenols (+10%), antioxidant activity (+14%), and color intensity (+10%), particularly when leaf removal was performed on the southern side. In addition, leaf removal increased free anthocyanins by 40% when applied to the south side of the canopy, 24% when applied to the north side, and 21% when applied to both the north and south sides. On the north, north–south, and south sides of the canopy, volatile chemicals were reduced by about 18, 14, and 13%, respectively, when the treatment was applied [26].

Zhang et al. [27] evaluated the impact of apical and basal defoliation on canopy structural parameters using photography of the canopy cover and computer vision methods.

During two harvests (2010–2011 and 2015–2016) in Yarra Valley, Australia, the impact of canopy structural changes on the chemical contents of grapes and wines was studied. Five distinct treatments were applied to the Shiraz grapevines: no leaf removal (Control) and basal (TB) and apical (TD) leaf removal at veraison and intermediate ripeness, respectively. The removal of basal leaves considerably decreased the leaf area index and foliage cover and increased the canopy porosity but the removal of apical leaves had no effects on the canopy metrics. Nonetheless, the latter often resulted in a wine with a lower alcohol content. There were statistically significant increases in pH and reductions in TA in shaded grapes but there were no significant changes in the wine's color profile or volatile components. These findings indicate that apical leaf removal is an efficient technique for reducing wine alcohol content with little effect on wine composition.

The quality of wines depends largely on the composition of grape berries from which they were produced. Faster ripening may mean higher alcohol and less developed aromas, so it may be necessary to slow down ripening. For the reasons outlined above, the solution from a viticultural point of view can be to reduce the leaf area. In this study, we aimed to delay ripening by reducing the canopy size by two different treatments: short topping and machine leaf removal in Badacsony, Hungary.

## 2. Materials and Methods

### 2.1. Experimental Site, Vineyard, and Growing Conditions

In Badacsony, we compared the results of small plots (10 vines) in 4 repetitions (40 vines) of both treated and control vines of 'Pinot noir' (early-red) and 'Welshriesling' (late-white), respectively. The selected vineyards were from the Hungarian University of Agricultural and Life Sciences, Institute for Viticulture and Oenology, Badacsony Research Station, and the results of plantations of all of the varieties of 0.2 ha for 'Welshriesling' and 0.3 ha for 'Pinot noir' in the same area. All the vines studied had with the same $2 \, \text{m} \times 1 \, \text{m}$ vine spacing ($0.5 \, \text{vines/m}^2$) and Teleki 5C (E20) rootstock and cordon training system. The bud load was set at $7 \, \text{buds/m}^2$ 14 buds/stock: 12-budded canes and 2 budded spurs during pruning.

### 2.2. Treatments

For the period 2019–2022, the following treatments were set at veraison for both of the varieties:

- LF (leaf removal): the leaves above the cluster zone were removed with a special leaf stripper at veraison (Figure 4);
- ST (short topping): the shoots were trimmed short (60–70 cm) at veraison;
- Control: no treatment was done.

The harvesting date was the same for all treatments but was dependent on the year and cultivar: the exact date was determined by sampling for both varieties in each experimental year. Based on the measurements, the harvest date was set so that the KMW reached 20 in the control plots.

### 2.3. Experimental Harvest Measures

Throughout the experimental harvests, the following parameters were determined: yield ($\text{kg/m}^2$), sugar content of the juice (Klosterneuburger Mostwaage = KMW g/100 g), titratable acidity of must (g/L), and pH (measured by electronic pH meter). The degree of rot (*Botrytis cinerea* infection %) was estimated visually.

### 2.4. Data Analyses

The homogeneity of variances and the distribution of the harvest results data (normality test) were checked by the Levene test and Shapiro–Wilk test, respectively, and then, as these do not meet the basic conditions for standard ANOVA, data were evaluated by Aligned Rank Transformed ANOVA [28] by 3 factors: treatment (LR, ST, and control):year (2019–2022) and cultivar (Pinot noir, Welshriesling). Where the ART–ANOVA

results indicated that the expected values differed at a significance level of at least 90%, the expected values were compared pairwise using the "Aligned Ranked Transform Contrasts" test [28,29]. All of the results were analysed and evaluated using the R software package [30]. The graphs were conducted using the ggplot2 package [31].

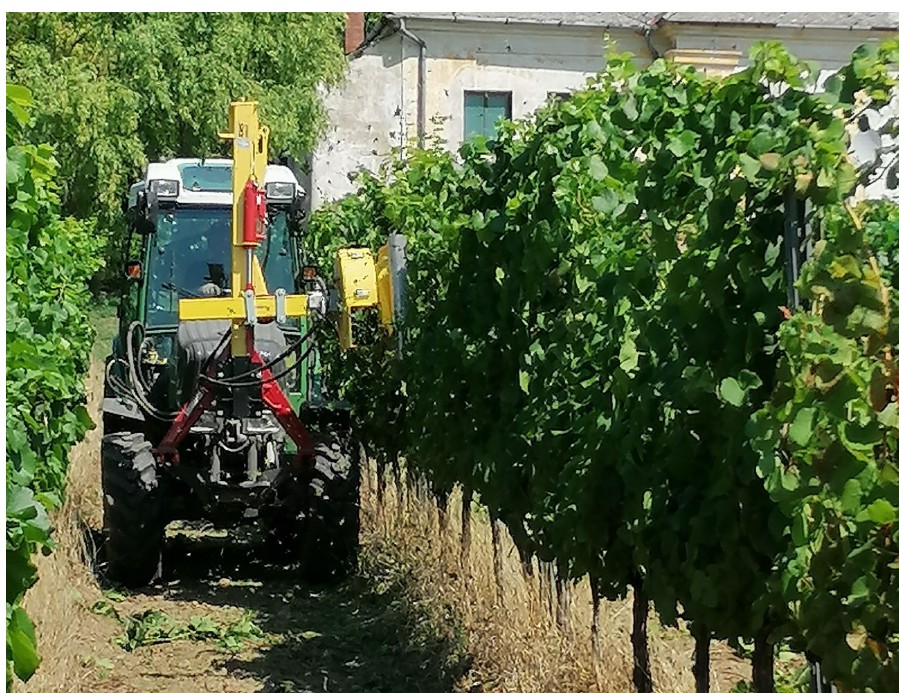

**Figure 4.** Mechanical leaf removal by tractor-mounted special leaf stripper machine.

## 3. Results

The primary objective of our experiment was to delay ripening by reducing the assimilation surface area and by reducing the sugar content of the grape juice, while keeping the yield and other harvest parameters unchanged. The results are presented below for each of the harvest parameters measured.

### 3.1. Yield

The yields measured during harvest are summarized in Table 1.

**Table 1.** Yields by cultivar and treatment (2019–2022, Badacsony, Hungary; data in kg/m$^2$).

| Cultivar | Pinot Noir | | | Welschriesling | | | Yearly Statistics |
|---|---|---|---|---|---|---|---|
| Year | Control | Short Topping | Leaf Removal | Control | Short Topping | Leaf Removal | |
| 2019 | 1.10 | 1.18 | 1.20 | 1.42 | 1.51 | 1.42 | |
| | 1.45 | 1.26 | 1.17 | 1.44 | 1.55 | 1.51 | |
| | 1.24 | 1.18 | 1.27 | 1.40 | 1.36 | 1.64 | |
| | 1.37 | 1.13 | 1.39 | 1.50 | 1.39 | 1.54 | |
| Average | 1.29 | 1.19 | 1.26 | 1.44 | 1.45 | 1.53 | 1.36 |
| Variance | 0.0235 | 0.0029 | 0.0096 | 0.0019 | 0.0084 | 0.0082 | 0.0224 |
| 2020 | 0.91 | 0.78 | 0.82 | 1.2 | 1.2 | 1.25 | |
| | 0.76 | 0.88 | 0.88 | 1.19 | 1.23 | 1.34 | |
| | 0.77 | 0.84 | 0.79 | 1.14 | 1.1 | 1.13 | |
| | 0.88 | 0.81 | 0.87 | 1.02 | 1.17 | 1.23 | |
| Average | 0.83 | 0.8275 | 0.84 | 1.1375 | 1.175 | 1.2375 | 1.01 |
| Variance | 0.0058 | 0.0018 | 0.0018 | 0.0068 | 0.0031 | 0.0074 | 0.0365 |

**Table 1.** *Cont.*

| Cultivar | Pinot Noir | | | Welschriesling | | | Yearly Statistics |
|---|---|---|---|---|---|---|---|
| Year | Control | Short Topping | Leaf Removal | Control | Short Topping | Leaf Removal | |
| 2021 | 0.81 | 0.63 | 0.42 | 1.67 | 1.57 | 1.68 | |
| | 0.82 | 0.44 | 0.56 | 1.51 | 1.24 | 1.09 | |
| | 0.96 | 0.63 | 0.47 | 1.41 | 1.5 | 1.88 | |
| | 0.74 | 0.82 | 0.69 | 1.81 | 1.16 | 1.39 | |
| Average | 0.8325 | 0.63 | 0.535 | 1.6 | 1.3675 | 1.51 | 1.08 |
| Variance | 0.0085 | 0.0241 | 0.0140 | 0.0311 | 0.0393 | 0.1189 | 0.2218 |
| 2022 | 1.15 | 1.20 | 1.22 | 1.8 | 1.89 | 1.75 | |
| | 1.14 | 1.28 | 1.21 | 1.85 | 1.82 | 1.71 | |
| | 1.24 | 1.19 | 1.22 | 1.61 | 1.69 | 1.80 | |
| | 1.11 | 1.20 | 1.20 | 1.71 | 1.60 | 1.72 | |
| Average | 1.16 | 1.2175 | 1.2125 | 1.7425 | 1.75 | 1.745 | 1.47 |
| Variance | 0.0031 | 0.0018 | 0.0001 | 0.0112 | 0.0169 | 0.0016 | 0.0836 |
| Average | | 0.99 | | | 1.47 | | |
| Variance | | 0.07 | | | 0.06 | | |

The highest yield was measured in 2022 (1.47 kg/m$^2$) and the lowest in 2020 (1.01 kg/m$^2$). On average, over four years, Welschriesling yielded one and a half times more than Pinot noir (1.47 and 0.99 kg/m$^2$, respectively). The ART–ANOVA analysis showed that while there were no significant differences between treatments in terms of the yield, there were significant differences between years (Figure 5a) and cultivars (Figure 5b).

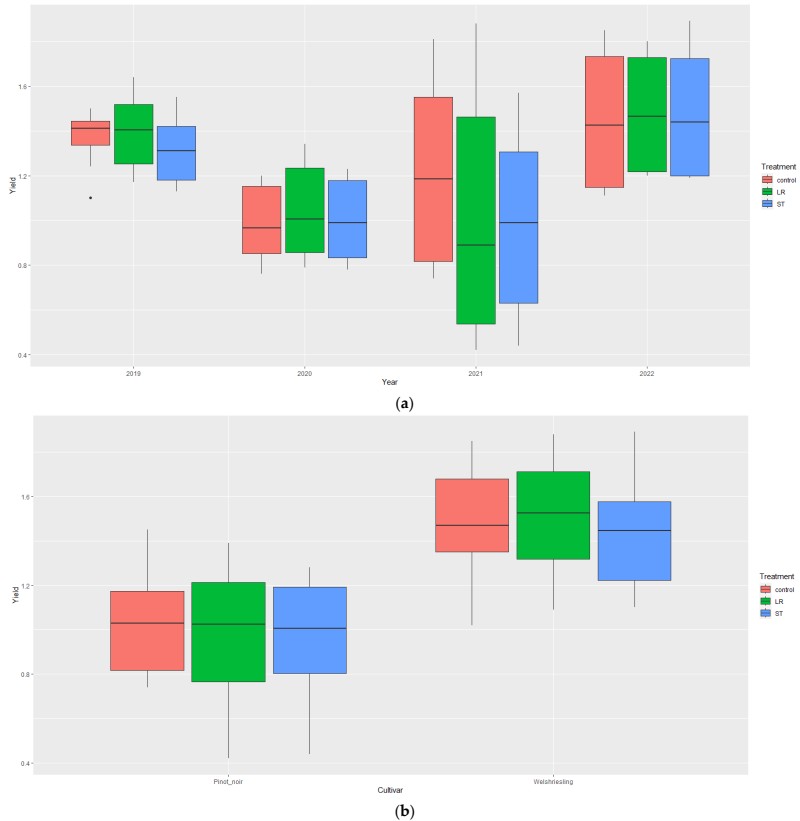

**Figure 5.** Effect of different treatments (LF = Leaf Removal; ST = Short Topping) on yield in different (**a**) years and (**b**) cultivars.

### 3.2. Sugar Content of Grape Juice

The sugar content of the grape juice was measured in Klosterneuburger Mostwaage (KMW g/100 g). The results are summarized in Table 2.

**Table 2.** Sugar content of grape juice by cultivar and treatment (2019–2022, Badacsony, Hungary; data in Klosterneuburger Mostwaage).

| Cultivar | Pinot Noir | | | Welshriesling | | | Yearly Statistics |
| --- | --- | --- | --- | --- | --- | --- | --- |
| Year | Control | Leaf Removal | Short Topping | Control | Leaf Removal | Short Topping | |
| 2019 | 19.80 | 18.20 | 18.70 | 19.30 | 21.30 | 21.90 | |
| | 20.90 | 18.40 | 19.40 | 22.70 | 21.10 | 22.50 | |
| | 19.40 | 17.90 | 17.50 | 24.00 | 19.10 | 19.50 | |
| | 19.50 | 18.00 | 18.40 | 23.90 | 21.00 | 19.30 | |
| Average | 19.90 | 18.13 | 18.50 | 22.48 | 20.63 | 20.80 | 20.07 |
| Variance | 0.4733 | 0.0492 | 0.6200 | 4.8292 | 1.0492 | 2.6800 | 3.5091 |
| 2020 | 19.60 | 18.70 | 17.80 | 21.10 | 20.10 | 20.10 | |
| | 19.40 | 18.80 | 19.20 | 21.40 | 20.00 | 20.40 | |
| | 19.20 | 18.00 | 18.40 | 20.90 | 20.50 | 20.10 | |
| | 18.80 | 18.80 | 17.40 | 21.30 | 19.90 | 20.30 | |
| Average | 19.25 | 18.58 | 18.20 | 21.18 | 20.13 | 20.23 | 19.59 |
| Variance | 0.1167 | 0.1492 | 0.6133 | 0.0492 | 0.0692 | 0.0225 | 1.2251 |
| 2021 | 22.70 | 20.70 | 22.40 | 22.50 | 22.10 | 22.50 | |
| | 23.10 | 21.70 | 21.50 | 22.90 | 21.10 | 22.90 | |
| | 21.50 | 19.80 | 21.70 | 22.60 | 22.20 | 21.50 | |
| | 21.90 | 22.20 | 21.60 | 22.70 | 21.50 | 21.70 | |
| Average | 22.30 | 21.10 | 21.80 | 22.68 | 21.73 | 22.15 | 21.96 |
| Variance | 0.53 | 1.14 | 0.17 | 0.03 | 0.27 | 0.44 | 0.5938 |
| 2022 | 21.50 | 19.20 | 19.60 | 19.80 | 19.80 | 20.40 | |
| | 21.30 | 18.50 | 20.30 | 20.90 | 20.30 | 21.70 | |
| | 21.40 | 18.40 | 20.20 | 22.00 | 20.50 | 18.70 | |
| | 19.70 | 21.20 | 19.50 | 20.80 | 20.70 | 18.90 | |
| Average | 20.98 | 19.33 | 19.90 | 20.88 | 20.33 | 19.93 | 20.22 |
| Variance | 0.73 | 1.69 | 0.17 | 0.81 | 0.15 | 1.98 | 1.0678 |
| Average | | 19.83 | | | 21.09 | | |
| Variance | | 2.3191 | | | 1.6348 | | |

In terms of the sugar content of grape juice, the 'Welshriesling' showed a higher value (21.09 Kl°) on average over four years while in 2021 the highest result (21.96 Kl°) and in 2020 the lowest (19.59 Kl°) values were detected.

The treatments reduced the sugar content of grape juice in both cultivars (Figure 6b). The sugar content of grape juice decreased in all years but the difference was only significant at the 99% level (alpha = 0.0.1) in 2019 (Figure 6a).

There was no statistically proven difference between the two treatments in either year or for either cultivar.

### 3.3. Titratable Acidity of Grape Juice

The results of the titratable acid content of grape juice are summarized in Table 3.

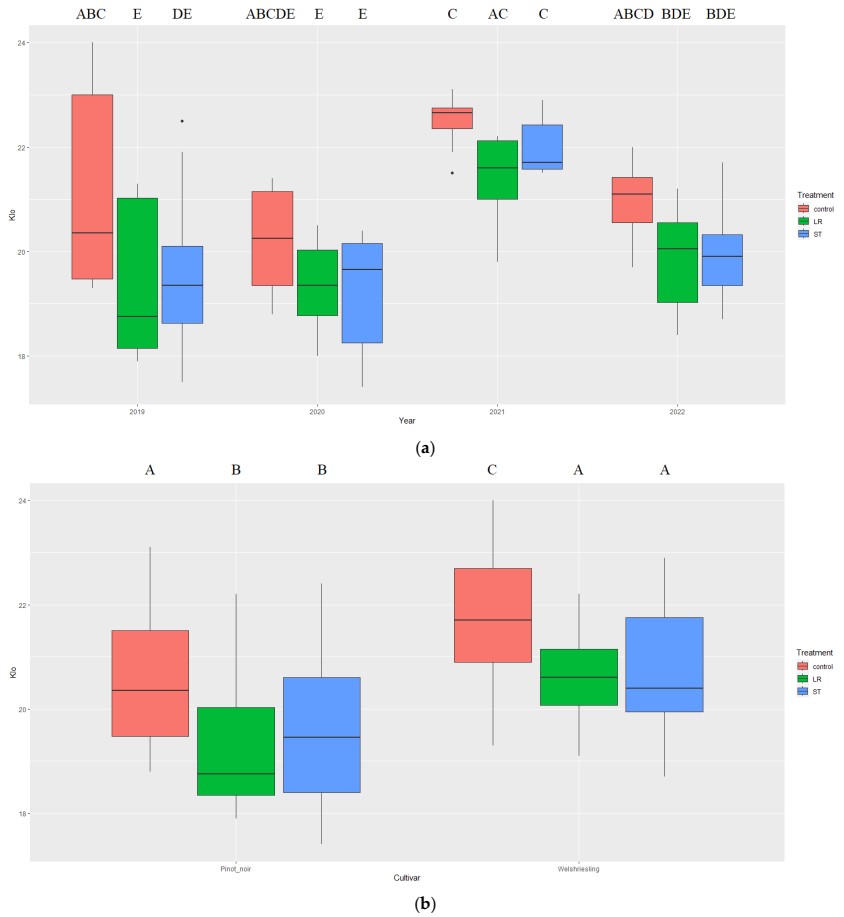

**Figure 6.** Effect of different treatments (LF = Leaf Removal; ST = Short Topping) on the sugar content of the grape juice in different (**a**) years and (**b**) cultivars. Different capital letter indicate a statistically significant difference.

**Table 3.** The titratable acid content of grape juice by cultivar and treatment (2019–2022, Badacsony, Hungary; data in g/L).

| Cultivar | Pinot Noir | | | Welshriesling | | | Yearly Statistics |
|---|---|---|---|---|---|---|---|
| Year | Control | Leaf Removal | Short Topping | Control | Leaf Removal | Short Topping | |
| 2019 | 8.50 | 7.26 | 7.63 | 6.89 | 7.33 | 5.09 | |
| | 9.54 | 7.25 | 6.82 | 6.06 | 6.22 | 7.11 | |
| | 8.50 | 7.20 | 9.12 | 7.65 | 7.72 | 7.51 | |
| | 8.74 | 8.41 | 8.09 | 6.17 | 6.6 | 6.64 | |
| Average | 8.82 | 7.53 | 7.92 | 6.69 | 6.97 | 6.59 | 7.42 |
| Variance | 0.2432 | 0.3449 | 0.9210 | 0.5430 | 0.4638 | 1.1231 | 1.1084 |
| 2020 | 10.89 | 9.44 | 9.35 | 7.14 | 7.12 | 7.02 | |
| | 8.78 | 9.06 | 8.27 | 6.78 | 6.79 | 6.79 | |
| | 10.37 | 8.53 | 8.11 | 7.03 | 7.37 | 7.53 | |
| | 10.53 | 9.09 | 9.58 | 8.05 | 7.14 | 7.2 | |
| Average | 10.14 | 9.03 | 8.83 | 7.25 | 7.11 | 7.14 | 8.25 |
| Variance | 0.8724 | 0.1409 | 0.5550 | 0.3071 | 0.0570 | 0.0975 | 1.6696 |

**Table 3.** *Cont.*

| Cultivar | Pinot Noir | | | Welshriesling | | | Yearly Statistics |
|---|---|---|---|---|---|---|---|
| Year | Control | Leaf Removal | Short Topping | Control | Leaf Removal | Short Topping | |
| 2021 | 9.06 | 10.5 | 9.68 | 4.26 | 4.24 | 4.59 | |
| | 8.89 | 8.05 | 9.54 | 4.43 | 4.69 | 5.5 | |
| | 8.3 | 8.4 | 8.55 | 5.39 | 5.78 | 6.23 | |
| | 9.84 | 8.05 | 7.82 | 4.27 | 6.91 | 5.07 | |
| Average | 9.02 | 8.75 | 8.90 | 4.59 | 5.41 | 5.35 | 7.00 |
| Variance | 0.4031 | 1.3883 | 0.7690 | 0.2923 | 1.4247 | 0.4843 | 4.4208 |
| 2022 | 6.84 | 6.60 | 6.97 | 4.94 | 5.56 | 5.78 | |
| | 6.8 | 6.71 | 7.00 | 4.84 | 5.26 | 5.05 | |
| | 6.63 | 6.10 | 6.04 | 5.33 | 5.03 | 5.46 | |
| | 6.79 | 6.30 | 6.73 | 5.28 | 5.05 | 5.46 | |
| Average | 6.77 | 6.43 | 6.69 | 5.10 | 5.23 | 5.44 | 5.94 |
| Variance | 0.0086 | 0.0777 | 0.1995 | 0.0595 | 0.0607 | 0.0895 | 0.5771 |
| Average | | 8.23 | | | 6.07 | | |
| Variance | | 1.6307 | | | 1.1942 | | |

Pinot noir had a significantly higher (8.23 g/L) acidity as compared to Welshriesling (6.07 g/L).

Only variety and vintage had a significant effect on the titratable acidity of grape juice but the cultivar:treatment interaction was also significant at the 99% level. This means that while there was no significant difference between treatments in terms of the acidity in Welshriesling, leaf removal significantly reduced it in Pinot noir (Figure 7).

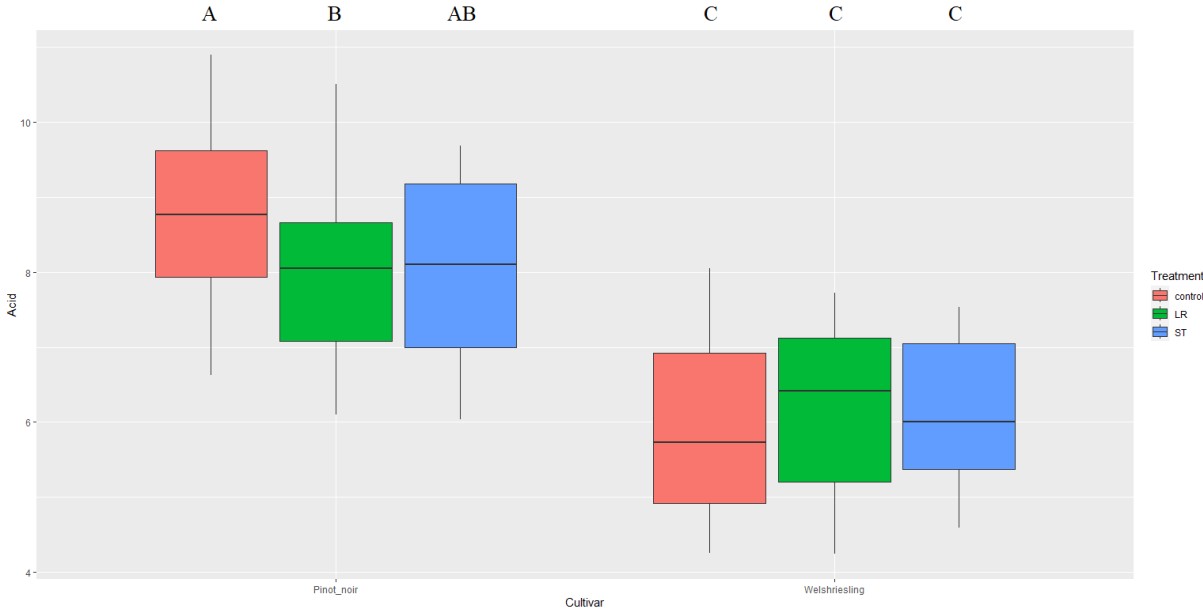

**Figure 7.** Effect of different treatments (LF = Leaf Removal; ST = Short Topping) on titratable acid content of the grape juice in different cultivars (data in g/L). Different capital letter indicate a statistically significant difference.

### 3.4. The pH Value of the Grape Juice

The pH value of the grape juice is determined each year for each treatment and cultivar as it is an important parameter in determining the acidity of the must and the wine made from it (Table 4).

**Table 4.** The pH value of grape juice by cultivar and treatment (2019–2022, Badacsony, Hungary).

| Cultivar | Pinot Noir | | | Welshriesling | | | Yearly Statistics |
|---|---|---|---|---|---|---|---|
| Year | Control | Leaf Removal | Short Topping | Control | Leaf Removal | Short Topping | |
| 2019 | 3.31 | 3.20 | 3.30 | 3.29 | 3.28 | 3.21 | |
| | 3.36 | 3.21 | 3.34 | 3.3 | 3.32 | 3.23 | |
| | 3.27 | 3.10 | 3.32 | 3.4 | 3.12 | 3.28 | |
| | 3.24 | 3.08 | 3.31 | 3.27 | 3.2 | 3.23 | |
| Average | 3.30 | 3.15 | 3.32 | 3.32 | 3.23 | 3.24 | 3.26 |
| Variance | 0.0027 | 0.0045 | 0.0003 | 0.0034 | 0.0079 | 0.0009 | 0.0063 |
| 2020 | 3.27 | 3.4 | 3.29 | 3.42 | 3.26 | 3.49 | |
| | 3.28 | 3.3 | 3.27 | 3.41 | 3.34 | 3.45 | |
| | 3.22 | 3.18 | 3.24 | 3.37 | 3.19 | 3.54 | |
| | 3.26 | 3.28 | 3.24 | 3.32 | 3.21 | 3.52 | |
| Average | 3.26 | 3.29 | 3.26 | 3.38 | 3.25 | 3.50 | 3.32 |
| Variance | 0.0007 | 0.0081 | 0.0006 | 0.0021 | 0.0045 | 0.0015 | 0.0108 |
| 2021 | 3.53 | 3.32 | 3.31 | 3.21 | 3.26 | 3.46 | |
| | 3.46 | 3.29 | 3.32 | 3.4 | 3.3 | 3.46 | |
| | 3.44 | 3.22 | 3.28 | 3.33 | 3.25 | 3.56 | |
| | 3.29 | 3.25 | 3.36 | 3.52 | 3.22 | 3.46 | |
| Average | 3.43 | 3.27 | 3.32 | 3.37 | 3.26 | 3.49 | 3.35 |
| Variance | 0.0102 | 0.0019 | 0.0011 | 0.0168 | 0.0011 | 0.0025 | 0.0115 |
| 2022 | 3.32 | 3.24 | 3.28 | 3.34 | 3.13 | 3.53 | |
| | 3.44 | 3.35 | 3.28 | 3.32 | 3.3 | 3.44 | |
| | 3.25 | 3.31 | 3.34 | 3.17 | 3.23 | 3.51 | |
| | 3.22 | 3.28 | 3.24 | 3.4 | 3.21 | 3.57 | |
| Average | 3.31 | 3.30 | 3.29 | 3.31 | 3.22 | 3.51 | 3.32 |
| Variance | 0.0096 | 0.0022 | 0.0017 | 0.0096 | 0.0049 | 0.0030 | 0.0127 |
| Average | 3.29 | | | 3.34 | | | |
| Variance | 0.0065 | | | 0.0150 | | | |

The grape cultivar had a high significance effect on the pH of the grape juice: Pinot noir had a lower pH (3.29) while Welshriesling had a higher pH (3.34).

The effect of treatments on the pH was significant at the 95% level (alpha = 0.05). Although the effect of neither treatment was significant compared to the control, the two treatments, namely short topping (ST) and leaf removal (LR), were significantly different, the first giving a lower and the second a higher value (Figure 8).

Looking at the effect of the treatments in different vintages, this difference was very marked in the 2019 vintage (Figure 9).

### 3.5. Rate of Botrytis Infection

The quality of a grape crop is influenced not only by the content of berries but also by its health. With this in mind, the rate of Botrytis infection of the grapes during the experimental harvest was also taken into account (Table 5).

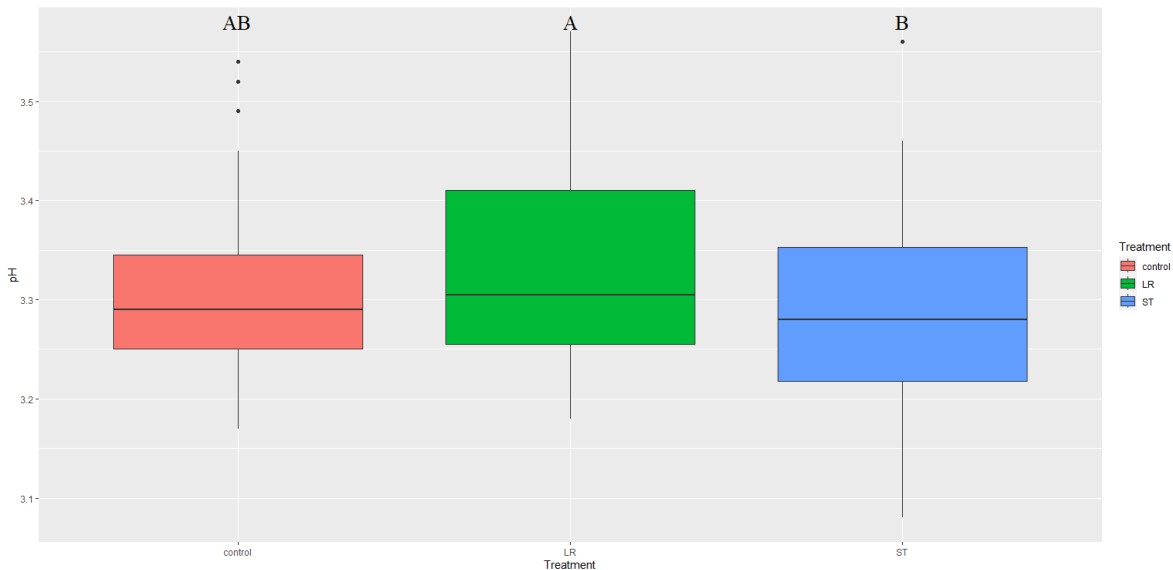

**Figure 8.** Effect of different treatments (LF = Leaf Removal; ST = Short Topping) on the pH value of the grape juice. Different capital letter indicate a statistically significant difference.

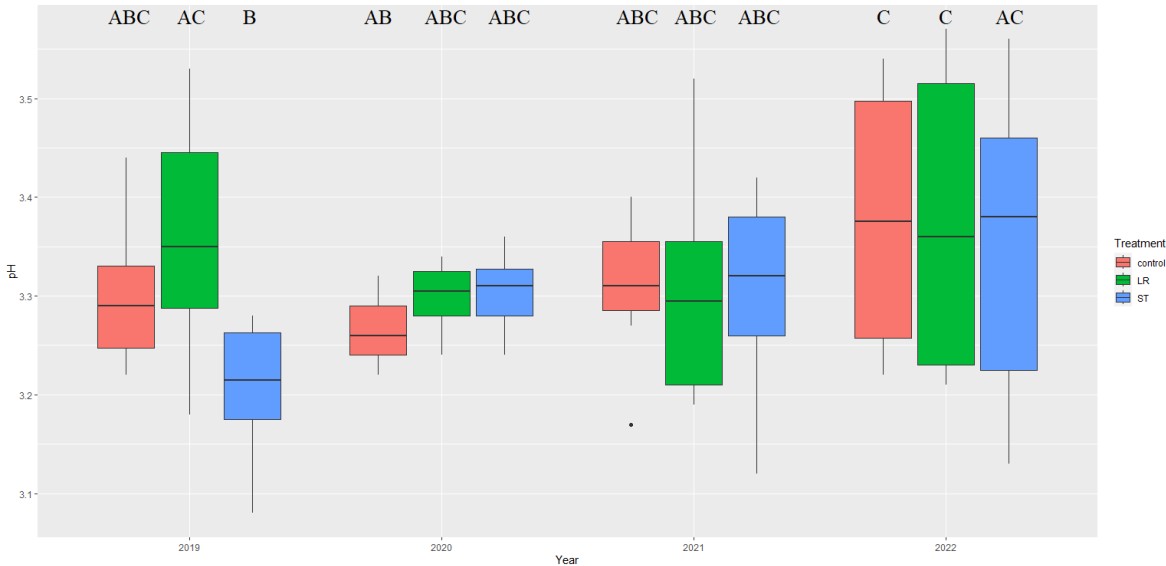

**Figure 9.** Effect of different treatments (LF = Leaf Removal; ST = Short Topping) on the pH of the grape juice in different years. Different capital letter indicate a statistically significant difference.

**Table 5.** Rate of Botrytis infection by cultivar and treatment (2019–2022, Badacsony, Hungary).

| Cultivar | Pinot Noir | | | Welshriesling | | | Yearly Statistics |
|---|---|---|---|---|---|---|---|
| Year | Control | Leaf Removal | Short Topping | Control | Leaf Removal | Short Topping | |
| 2019 | 0.00 | 5.00 | 10.00 | 0.00 | 5.00 | 0.00 | |
| | 0.00 | 5.00 | 7.00 | 0.00 | 5.00 | 0.00 | |
| | 0.00 | 5.00 | 5.00 | 0.00 | 5.00 | 0.00 | |
| | 0.00 | 5.00 | 5.00 | 0.00 | 5.00 | 0.00 | |
| Average | 0.00 | 5.00 | 6.75 | 0.00 | 5.00 | 0.00 | 2.79 |
| Variance | 0.0000 | 0.0000 | 5.5833 | 0.0000 | 0.0000 | 0.0000 | 9.2156 |

**Table 5.** *Cont.*

| Cultivar | Pinot Noir | | | Welshriesling | | | Yearly Statistics |
|---|---|---|---|---|---|---|---|
| Year | Control | Leaf Removal | Short Topping | Control | Leaf Removal | Short Topping | |
| 2020 | 2.00 | 15.00 | 5.00 | 0.00 | 5.00 | 0.00 | |
| | 3.00 | 15.00 | 0.00 | 0.00 | 10.00 | 0.00 | |
| | 3.00 | 15.00 | 5.00 | 0.00 | 5.00 | 0.00 | |
| | 3.00 | 15.00 | 2.00 | 0.00 | 10.00 | 0.00 | |
| Average | 2.75 | 15.00 | 3.00 | 0.00 | 7.50 | 0.00 | 4.71 |
| Variance | 0.2500 | 0.0000 | 6.0000 | 0.0000 | 8.3333 | 0.0000 | 30.5634 |
| 2021 | 0.00 | 10.00 | 3.00 | 0.00 | 0.00 | 0.00 | |
| | 0.00 | 10.00 | 5.00 | 0.00 | 0.00 | 0.00 | |
| | 0.00 | 10.00 | 3.00 | 0.00 | 0.00 | 0.00 | |
| | 0.00 | 10.00 | 0.00 | 0.00 | 0.00 | 0.00 | |
| Average | 0.00 | 10.00 | 2.75 | 0.00 | 0.00 | 0.00 | 2.13 |
| Variance | 0.0000 | 0.0000 | 4.2500 | 0.0000 | 0.0000 | 0.0000 | 14.5489 |
| 2022 | 10.00 | 5.00 | 2.00 | 5.00 | 0.00 | 0.00 | |
| | 10.00 | 10.00 | 2.00 | 10.00 | 0.00 | 0.00 | |
| | 10.00 | 5.00 | 3.00 | 10.00 | 0.00 | 0.00 | |
| | 10.00 | 5.00 | 2.00 | 5.00 | 0.00 | 0.00 | |
| Average | 10.00 | 6.25 | 2.25 | 7.50 | 0.00 | 0.00 | 4.33 |
| Variance | 0.0000 | 6.2500 | 0.2500 | 8.3333 | 0.0000 | 0.0000 | 17.1884 |
| Average | | 5.31 | | | 1.67 | | |
| Variance | | 20.6024 | | | 9.9291 | | |

Statistical analyses have produced contradictory results.

When looking at both cultivars in all years, short topping (ST) significantly increased the level of Botrytis infection.

The varieties were significantly affected by Botrytis infection rates, with Pinot noir less infected (Figure 10).

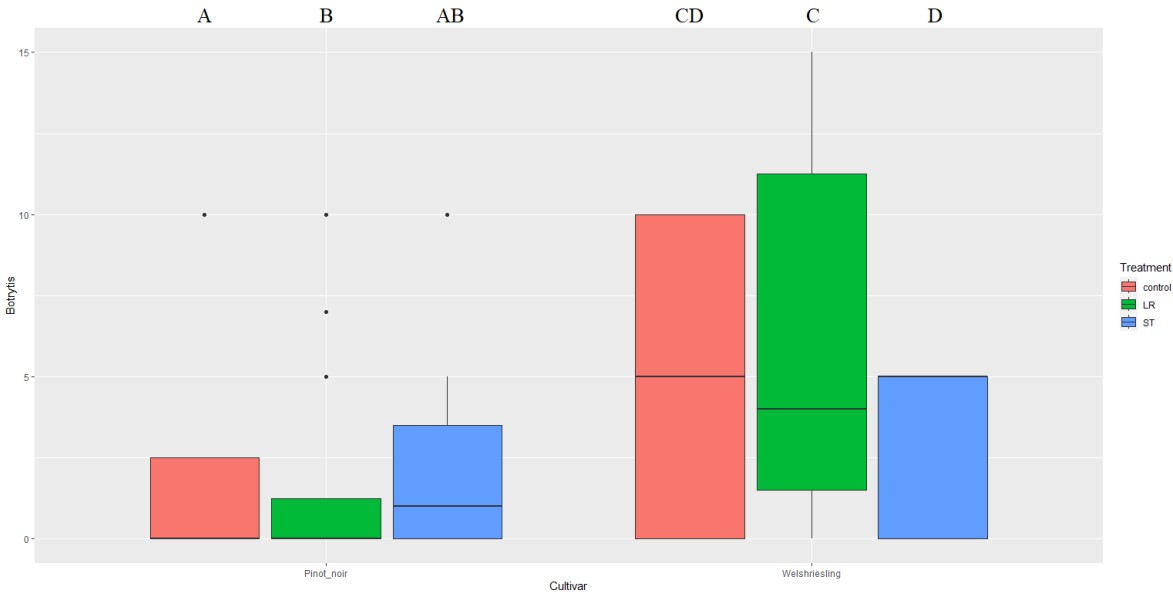

**Figure 10.** Effect of different treatments (LF = Leaf Removal; ST = Short Topping) on Botrytis infection of berries in different cultivars. Different capital letter indicate a statistically significant difference.

If we look at the results year by year, we can see that in 2019 there was only a difference between treatments, with short topping significantly reducing the rate of rot (Figure 11).

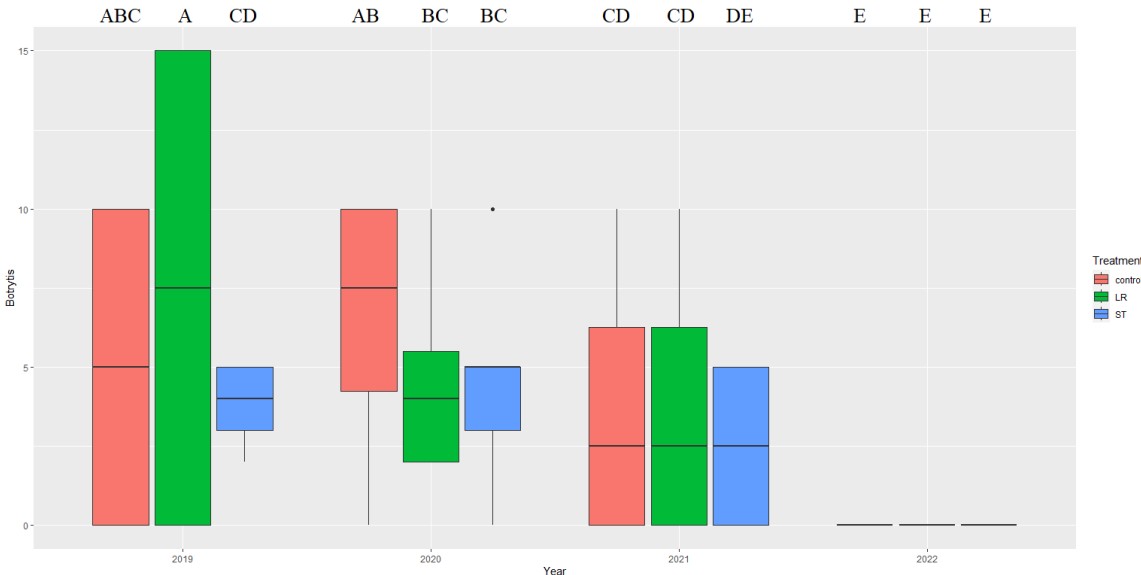

**Figure 11.** Effect of different treatments (LF = Leaf Removal; ST = Short Topping) on Botrytis infection of berries in different years. Different capital letter indicate a statistically significant difference.

### 3.6. Cultivar-Wise Analyses

The results of the cultivar-wise Aligned Rank Transformed ANOVA are presented in Table 6.

**Table 6.** Results of the cultivar-wise statistical analyses.

| Cultivar | Effect | Yield | Sugar Content of the Must | Titratable Acids | pH | Botrytis Infection |
|---|---|---|---|---|---|---|
| Pinot noir | Treatment | . | *** | ** | * | *** |
|  | Year | *** | *** | *** | *** | *** |
|  | Treatment:Year | ** |  |  | . | *** |
| Welshriesling | Treatment |  | *** |  | . | *** |
|  | Year | *** | *** | *** |  | *** |
|  | Treatment:Year |  |  |  | . | *** |

Significance codes: 0 '***' 0.001 '**' 0.01 '*' 0.05 '.' 0.1 ' ' 1.

#### 3.6.1. Pinot Noir

As shown in Table 6, all parameters of the Pinot noir cultivar were affected by the treatments, albeit at different levels of significance.

The yield was reduced by leaf removal (LR) at the 90% significance level (Figure 8) but the average reduction was only 6.5% (0.067 kg/m$^2$). The effect of vintage on the yield was more robust (99.9%) while the interaction between the year and treatment was 99% significant (Figure 12).

The effect of treatments on sugar content of the grape juice was significant at the 99.9% level, as it was also shown in Section 3.2.

The effect of treatments was significant at the 99% level, leaf removal decreased titratable acidity of the grape juice significantly compared to the control while there was no detectable difference between the short topping and the control in the case of Pinot noir (Figure 13).

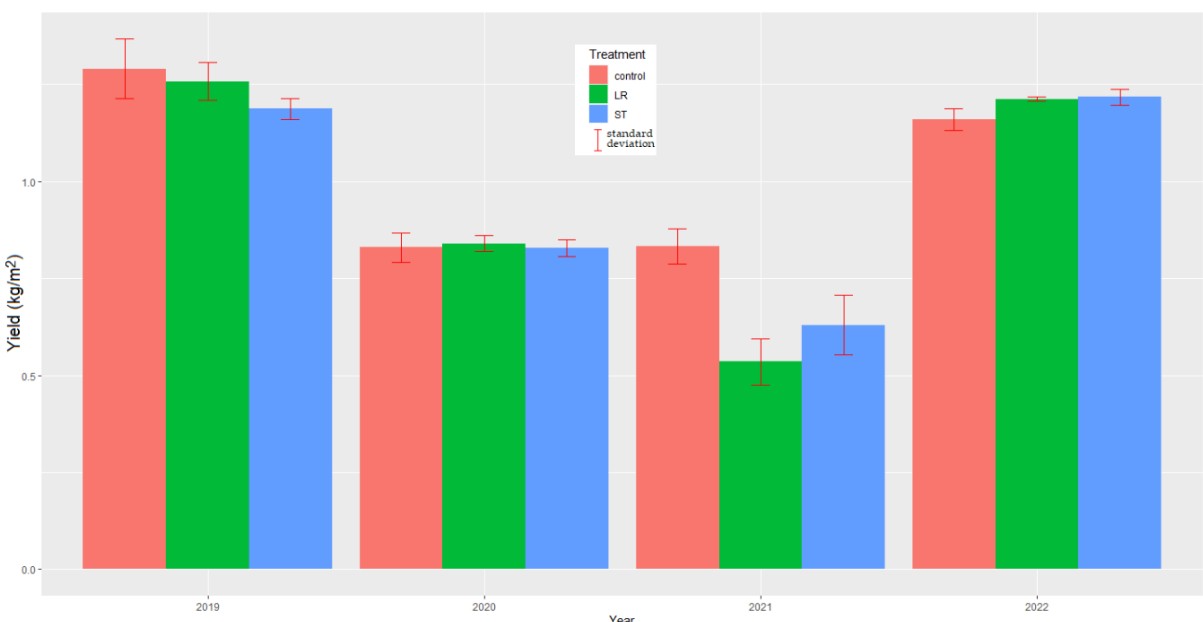

**Figure 12.** Effect of different treatments (LF = Leaf Removal; ST = Short Topping) on the Pinot noir yield in different years.

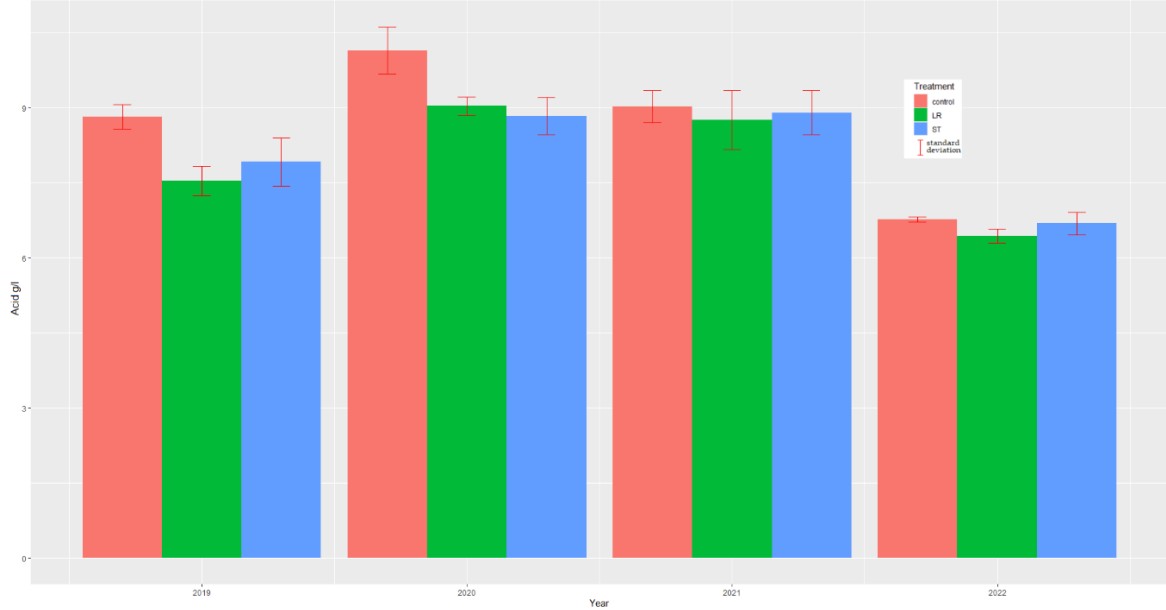

**Figure 13.** Effect of different treatments (LF = Leaf Removal; ST = Short Topping) on titratable acids of Pinot noir grape juice in different years (Badacsony, 2019–2022).

For Pinot noir, the treatment effect on pH was significant at the 99% level but so was the treatment:year interaction at the 95% level. As a result of the involvement in the interaction, the results may be ambiguous and a detailed post-hoc study was conducted. The results showed that only in year 2019 did the two treatments differ from each other; but neither differed from the control (Figure 14).

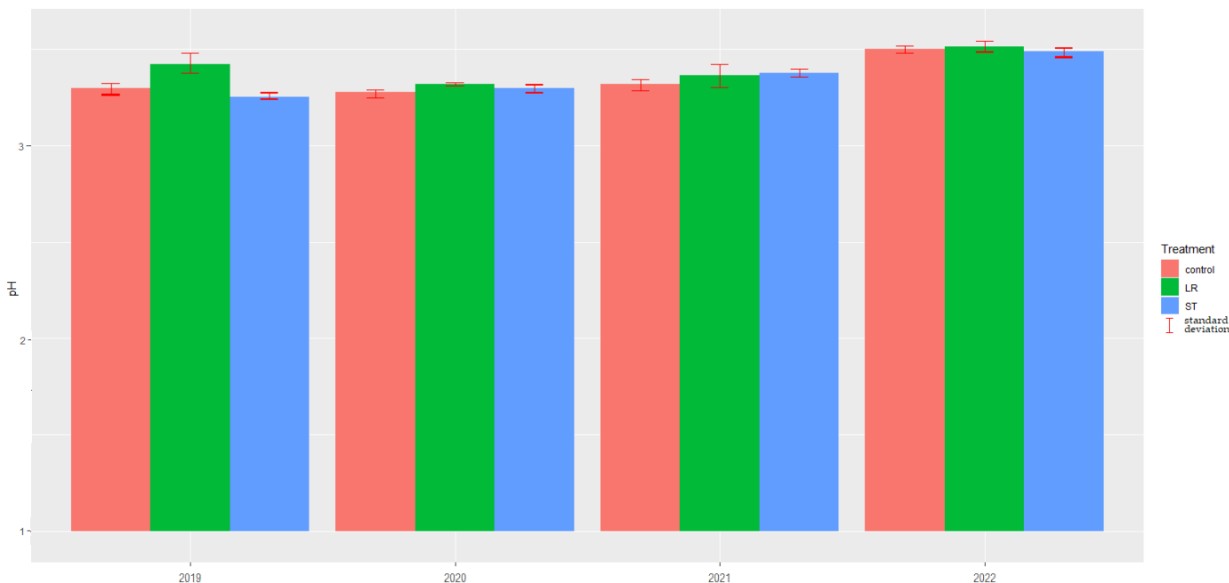

**Figure 14.** Effect of different treatments (LF = Leaf Removal; ST = Short Topping) on the pH of Pinot noir grape juice in different years (Badacsony, 2019–2022).

The effect of treatments on Botrytis infection of the grape berries was significant at the 99.9% level, as it was also shown in Section 3.5.

### 3.6.2. Welshriesling

For the Welshriesling variety, treatments had an effect on must sugar content and Botrytis infection at the 99.9% significance level; these factors have already been described in Sections 3.2 and 3.5.

In Welshriesling, the treatment effect on pH was significant at the 95% level but so was the treatment:year interaction. As the results could be misleading due to involvement in the interaction, we investigated what was causing the discrepancy. A detailed post-hoc showed that only the 2019 and 2020 short topping differed, which was clearly the year effect and not the treatment effect (Figure 15).

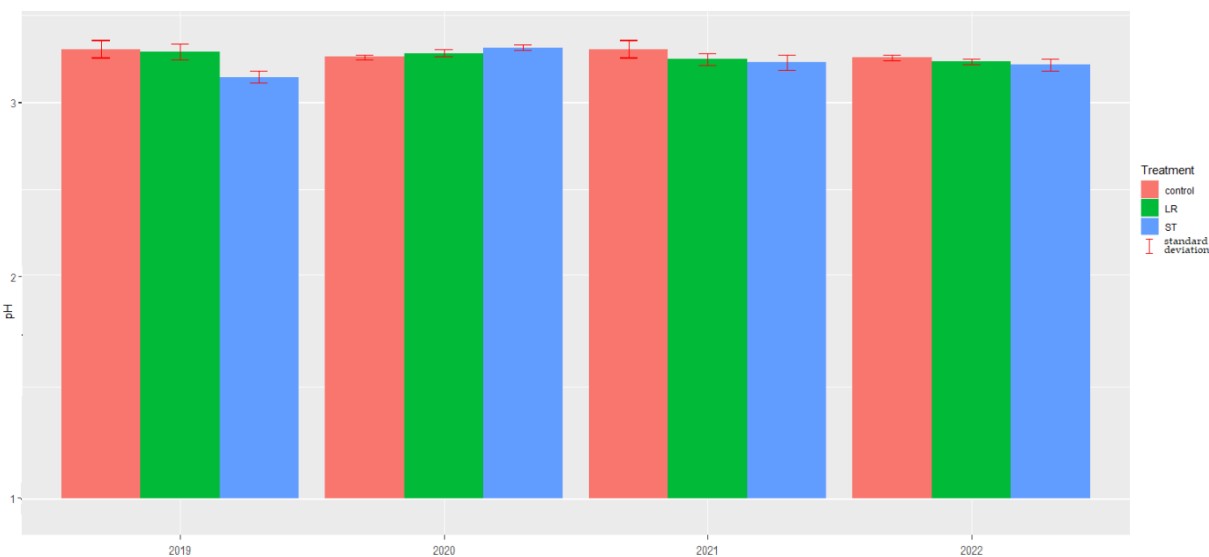

**Figure 15.** Effect of different treatments (LF = Leaf Removal; ST = Short Topping) on the pH of Welshriesling grape juice in different years (Badacsony, 2019–2022).

## 4. Discussion

The aim of our experiments was to slow down the ripening of the grapes so that we could achieve lower alcohol levels in the wines made from them. This is basically necessary because, although Hungary is close to the northern border of the grape-growing zone, climate change has increasingly caused the grapes to accumulate too much sugar due to rapid ripening, which has resulted in wines with disharmonious wines. Bringing the harvest date forward may offer a solution but it can have a negative impact on the acid composition and the development of the aromas responsible for the varietal character.

The architecture of the grape plant is intertwined with the procedures of training, formation, and pruning grape plants. These strategies set the conditions for espaliering the device that utilizes solar energy to form the organic mass of plants [32]. Our experiments were set up on an international early-ripening red (Pinot noir) and a regional late-ripening white (Welshriesling) cultivar. Our results showed that both treatments (short topping and leaf removal) were effective in reducing the sugar content of the grape juice. This effect was probably due to a lower level of photosynthesis in the treated vines than in the control vines due to a smaller assimilation surface.

High irradiation is not the only factor that leads to higher sugar content in berries. The cultivated grape (*Vitis vinifera* L.) is a C3 plant which means that it uses the Calvin cycle for atmospheric $CO_2$ fixation. At least three key issues inhibit the growth and production of C3 plants: high photorespiration (an unavoidable result of oxygenase activity of rubisco), a high water need, and a preference for temperate climates. As well as rubisco oxygenase activity, photorespiration was an adaptation to the current $CO_2/O_2$ levels in the atmosphere. Hence, the higher $CO_2$ might increase the photosynthetic efficiency and productivity of C3 plants [33] resulting in higher sugar accumulation.

Using an integrated model of canopy light interception, leaf thermal balancing, and photosynthetic processes, global maps of the theoretical maximums of grapevine canopy photosynthetic gain during berry development under current and future climatic scenarios were created. In future scenarios, the high-latitude zone accommodated high-gain sites typified by shifted appropriate regions and higher atmospheric $CO_2$ concentrations. In contrast, in a number of famous locations at low latitudes, the forecasted leaf temperatures surpassed the ideal range for photosynthesis, resulting in a decrease in gain [34].

Although Hungary falls within the ideal zone, traditional varieties and plantations are already experiencing the adverse effects of increased irradiation.

In a study conducted in Greece, temperature rises were found to have less of an effect on late-ripening cultivars than on international ones. Indigenous Greek varieties seem better suited to the region's recent and expected future climate, reacting less to warming than international cultivars in the majority of studied situations [35]. Similarly, the results reported in this study show that the 'Welshriesling', considered to be indigenous, was less affected by the treatments than the international Pinot noir.

The leaf area to fruit weight ratio (LA:FW) is often regarded as an essential factor in determining the overall performance of a vineyard [36,37]. In general, it is thought to be important to have a LA:FW ratio of at least 1 $m^2$/kg in order to provide optimal ripening conditions, in particular, sugar build-up [38]. Reduced LA:FW ratios may significantly slow down the veraison process and the buildup of sugar in grapes, although this has little influence on the overall acidity [39,40]. Similarly, in this study, the aim was to slow down ripening by changing the LA:FW ratio; here too the treatments were found to have little effect on the titratable acidity and pH.

In any case, our results showed that the acidity of Pinot noir grapes decreased, albeit slightly, but the leaf removal decreased the acidity and increased the pH. While acid loss is a serious problem for white wines, it is less of a problem for red wines as there is a difference between white and red wines when it comes to judging the acidity of it by the consumer. While white wines are generally expected to have a pronounced acidity [41], red wine drinkers tend to prefer softer wines.

Results that seem to contradict our findings have been reported in other studies: tests conducted with potted vines [42] indicated that the removal of leaves had only a temporary effect on vine physiology and had a little to non-existent impact on the grape berry composition. Similar results were found with field-grown vines [43]. It should be mentioned that in these experiments the rate of leaf removal was lower and the leaf area to crop weight ratio was more than 1 $m^2/kg$ in all treatments.

Depending on the year and the type of grapevine (white-Semillon, red-Shyraz), De Bei and co-authors [44] discovered that the influence of post-veraison leaf removal on phenology and grape composition was irregular. Despite this, the LA:FW ratio was higher than 1 $m^2/kg$ of fruit in all of the treatments.

## 5. Conclusions

Based on our results, we can conclude that both varieties and both treatments have been statistically proven to reduce the sugar content of the grapes, so, these methods can be successfully used in the future to delay ripening.

**Author Contributions:** Conceptualization, G.J. and B.Á.S.; methodology, G.K.S., C.N. and B.G.J.; validation, G.K.S., C.N. and B.G.J.; formal analysis, G.J.; investigation, G.J.; data curation, G.K.S.; writing—original draft preparation, G.J.; writing—review and editing, G.J., S.S. and Á.P.S.; visualization, G.J.; supervision, D.Á.N.S.; project administration, G.K.S. All authors have read and agreed to the published version of the manuscript.

**Funding:** This research received no external funding.

**Data Availability Statement:** Data is contained within the article.

**Conflicts of Interest:** The authors declare no conflict of interest.

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
