# Peer review of "Delay in the Ripening of Wine Grapes: Effects of Specific Phytotechnical Methods on Harvest Parameters"

_agronomy, doi:10.3390/agronomy13081963_

Round 1
Reviewer 1 Report
Dear authors,
the great effort has been made in this experiment, especially regarding four years long investigations.
But, overall merit is that we didn't get anything new from you. Reducing leaf area will probably reduce sugar and acidity accumulation. But what happened with other compounds like aromas or phenolics?
That would be of graeter interest to the readers and scientific community.
Beside that, manuscript lacks some basics informations. So, my decision would be reject, but try to publish in some less pronounced journal.
Here are some specific comments:
L 32. were only
L 182-190. delete
L 193. 10 grapevines or vines instead of stocks. So, how many grapevines per treatment?
L 195. University
L. 197. not clear
L 203. when?
L 205. when and how intensive?
L. 207 when?
L 216. How did you measure pH?
L 236. can we see yield/vine or at least number of vines per m2? In this way it is not comparable.
L 297-299. delete
Table 5. sort topping
L 398. short tapping
I don't have any specific comments.
Author Response
Dear Reviewer,
We would like to thank for reading and correcting our manuscript. According to your suggestions, we have rectified the shortcomings. We did our best in order to achieve the best results. We hope you find this modified version suitable for publication.
The great effort has been made in this experiment, especially regarding four years long investigations. But, overall merit is that we didn't get anything new from you. Reducing leaf area will probably reduce sugar and acidity accumulation. But what happened with other compounds like aromas or phenolics? That would be of graeter interest to the readers and scientific community.
This proposal is very interesting and forward-looking, but unfortunately in the current financial situation it is difficult for us to carry out major measurements, but we are looking for the possibility to carry out the measurements you suggest in the future in relation to our experiments in cultivation technology.
Beside that, manuscript lacks some basics informations. So, my decision would be reject, but try to publish in some less pronounced journal.
We're sorry you believe that.
Here are some specific comments:
L 32. were only
Corrected.
L 182-190. delete
Deleted.
L 193. 10 grapevines or vines instead of stocks. So, how many grapevines per treatment?
It has been corrected.
L 195. University
Corrected.
- 197. not clear
The sentence has been rephrased.
L 203. when?
Completed.
L 205. when and how intensive?
Completed.
- 207 when?
Completed.
L 216. How did you measure pH?
Completed.
L 236. can we see yield/vine or at least number of vines per m2? In this way it is not comparable.
In materials and methods, the vine spacing is indicated (2 x 1m), which means that there is 0.5 vines/m2. If you multiply the data by 2 in table 1, you get the results in kg/vine.
L 297-299. delete
It has been deleted.
Table 5. sort topping
Corrected.
L 398. short tapping
Corrected.

Reviewer 2 Report
In the example of two grape varieties grown in Hungary, the authors confirmed the practical effect of machine leaf removal and short topping on the delay of berry ripening and reduction of sugar content in them.
Excerpt from the study between lines 182 and 190: “The introduction should briefly place….. further details on references.” is redundant.
Abstract: Due to climate change, the sugar content of grapes in Hungary has increased so that the high alcohol content alone can make wines disharmonious. In most vintages, this phenomenon is only a problem for early-ripening varieties. In order to prevent and treat this difficulty, we have been carrying out experiments in grape canopy management for four years with the aim of delaying ripening and thus reducing the sugar content of the grapes. The experiments were set up on an early (Pinot noir) and a late (Welshriesling) variety; two treatments (leaf removal - LR and short topping - ST) were applied and compared to untreated control in the years 2019-2022. Our results showed that must most sugar yield was significantly reduced in all four years and for both of cultivars, while the other measured parameters (yield, acidity, pH, Botrytis infection) were not or lightly affected.
Author Response
Dear Reviewer,
We would like to thank for reading and correcting our manuscript. According to your suggestions, we have rectified the shortcomings. We did our best in order to achieve the best results. We hope you find this modified version suitable for publication.
In the example of two grape varieties grown in Hungary, the authors confirmed the practical effect of machine leaf removal and short topping on the delay of berry ripening and reduction of sugar content in them.
Excerpt from the study between lines 182 and 190: “The introduction should briefly place….. further details on references.” is redundant.
This part of the manuscript has been deleted.
The highlighted linguistic errors in the abstract text has been corrected.

Reviewer 3 Report
Some points were highlighted and commented in the MS.

The quality is good. some minor problems should be considered like scientific names in Italic
all highlighted through the MS.
Author Response
Dear Reviewer,
We would like to thank for reading and correcting our manuscript. According to your suggestions, we have rectified the shortcomings. We did our best in order to achieve the best results. We hope you find this modified version suitable for publication.
Some points were highlighted and commented in the MS.
All of the highlighted errors has been corrected.

Reviewer 4 Report
Dear editor,
I carefully reviewed the manuscript entitled as ‘Delay of the Ripening of Wine Grapes: Effects of Specific Phytotechnical Methods on Harvest Parameters’ wherein the effect of grape canopy management on yield and some biochemical traits of 2 grape genotypes. Although the findings of the manuscript are novel, poorly written article, it requires major modifications. I think this manuscript has the potential to be published in this journal, but some important points have to be clarified or fixed as follows:
o I think that this is not the final version of the Manuscript (see L182-190).
o The abstract should be rewritten. Some results are given in more detail in the abstract.
o The introduction is poorly structured and needs a major rewrite.
o Some cases must be explained by materials and methods, such as "degree of rot" measurement.
o Both tables and graphs are not well prepared. The results of tables and figures are confusing. Comparison of averages has not been done.
o The results must be better explained and justified. I recommend revising the discussion on your results.
o The conclusion section appears to be a summary. I recommend revising the conclusion based on your results.
Author Response
Dear Reviewer,
We would like to thank for reading and correcting our manuscript. According to your suggestions, we have rectified the shortcomings. We did our best in order to achieve the best results. We hope you find this modified version suitable for publication.
I carefully reviewed the manuscript entitled as ‘Delay of the Ripening of Wine Grapes: Effects of Specific Phytotechnical Methods on Harvest Parameters’ wherein the effect of grape canopy management on yield and some biochemical traits of 2 grape genotypes. Although the findings of the manuscript are novel, poorly written article, it requires major modifications. I think this manuscript has the potential to be published in this journal, but some important points have to be clarified or fixed as follows:
I think that this is not the final version of the Manuscript (see L182-190).
This paragraph has been deleted. Thank you for your notice.
The abstract should be rewritten. Some results are given in more detail in the abstract.
The abstract has been corrected.
The introduction is poorly structured and needs a major rewrite.
The introduction has been completed and reworded.
Some cases must be explained by materials and methods, such as "degree of rot" measurement.
Degree of rot was visually estimated. Unfortunately, due to financial difficulties we cannot use more adequate methods, but we have been using this method for many years and it has been published in several publications (e.g. in Scientia Horticulturae DOI: 10.1016/j.scienta.2022.111501).
Both tables and graphs are not well prepared. The results of tables and figures are confusing. Comparison of averages has not been done.
Obvious errors in tables and figures have been corrected. Averages were compared using ART-ANOVA, as detailed in the Materials and methods section.
The results must be better explained and justified. I recommend revising the discussion on your results.
The results and the discussion chapters have been revised.
The conclusion section appears to be a summary. I recommend revising the conclusion based on your results.
The conclusions have been revised.

Round 2
Reviewer 1 Report
Dear authors,
although you have made significant effort to improve your manuscript, my main concern (novelity and significance of content) still remains.
But, since you have made most of required changes, the manuscript could be accepted if the editors think that would be in line with journal policy.
None.
Reviewer 4 Report
I think the manuscript could be acceptable for publishing in Agronomy.